# Influence of Processing Route on the Surface Reactivity of Cu$_{47}$Ti$_{33}$Zr$_{11}$Ni$_6$Sn$_2$Si$_1$ Metallic Glass

Erika Soares Barreto [1,2,3,*], Volker Uhlenwinkel [1,2], Maximilian Frey [4], Isabella Gallino [4], Ralf Busch [4] and Andreas Lüttge [3]

1   Leibniz Institute for Materials Engineering, Badgasteiner Str. 3, 28359 Bremen, Germany; uhl@iwt.uni-bremen.de
2   Department of Production Engineering, Particle and Process Engineering, University of Bremen, Badgasteiner Str. 1-3, 28359 Bremen, Germany
3   Faculty of Geoscience, University of Bremen, Klagenfurter Str. 2-4, 28359 Bremen, Germany; aluttge@uni-bremen.de
4   Chair of Metallic Materials, Saarland University, Campus C6.3, 66123 Saarbrücken, Germany; maximilian.frey@uni-saarland.de (M.F.); i.gallino@mx.uni-saarland.de (I.G.); r.busch@mx.uni-saarland.de (R.B.)
*   Correspondence: sbarreto@iwt.uni-bremen.de; Tel.: +49-421-218-64514

**Abstract:** Recently, laser additive manufacturing (AM) techniques have emerged as a promising alternative for the synthesis of bulk metallic glasses (BMGs) with massively increased freedom in part size and geometry, thus extending their economic applicability of this material class. Nevertheless, porosity, compositional inhomogeneity, and crystallization display themselves to be the emerging challenges for this processing route. The impact of these "defects" on the surface reactivity and susceptibility to corrosion was seldom investigated but is critical for the further development of 3D-printed BMGs. This work compares the surface reactivity of cast and additively manufactured (via laser powder bed fusion—LPBF) Cu$_{47}$Ti$_{33}$Zr$_{11}$Ni$_6$Sn$_2$Si$_1$ metallic glass after 21 days of immersion in a corrosive HCl solution. The cast material presents lower oxygen content, homogeneous chemical distribution of the main elements, and the surface remains unaffected after the corrosion experimentation based on vertical scanning interferometry (VSI) investigation. On the contrary, the LPBF material presents a considerably higher reactivity seen through crack propagations on the surface. It exhibits higher oxygen content, heterogeneous chemical distribution, and presence of defects (porosity and cracks) generated during the manufacturing process.

**Keywords:** metallic glasses; surface reactivity; additive manufacturing; elemental mapping distribution; vertical scanning interferometry

## 1. Introduction

Since the first synthesis in the 1960s, metallic glasses have been extensively investigated, especially in terms of mechanical properties. The combination of high strength, superior hardness, large elastic strain limit, and improved wear and corrosion resistance make them promising candidates for a wide variety of engineering applications [1]. These properties are commonly attributed to the amorphous state, associated with the absence of grain boundaries, segregates, and microstructural defects.

In terms of corrosion behavior, most literature reports upon the influence of the chemical composition of the glass-forming alloys on corrosion kinetics [2–5]. Alloys with one early and one late transition element, e.g., Cu-Ti-based, have a great difference in electrochemical potentials of the constituents. This can lead to oxidation and partitioning effects, which in turn promotes crystallization due to the thermodynamical destabilization of the amorphous phase [4]. The corrosion resistance of metallic glasses can be improved by minor element additions by impacting either the kinetics or thermodynamics of reactions [3,4,6–8]. The atomic mobility can be reduced when small amounts of large-size elements are added [4].

Elements that form stable oxide layers, however, enhance the thermodynamics of repassivation. For many metallic glasses, the nature of corrosion resistance relates to the formation of a passive film that, when disrupted, may cause severe localized corrosion [9]. Moreover, the susceptibility to local corrosion is associated with the existence of defects and chemical inhomogeneities that can hardly be avoided during material synthesis [9].

Only little information has been reported regarding the relationship between surface reactivity of metallic glasses and the respectively applied processing route. Most of the reported studies, including the aforementioned literature, were performed on samples produced by conventional casting or melt spinning. Melt spinning is a rapidly quenching technique to form a glass, thus restricting the sample geometry to ribbons or foils of several micrometers in thickness. In contrast, mold casting results in slower cooling rates, therefore requiring advanced alloy compositions with increased glass-forming abilities (GFA). In this case, the glass is produced monolithically via heat transfer from the liquid into the mold, where the maximum cooling rate achievable in the inner regions of the mold cavity decreases with the increasing thickness of the part. This leads to an alloy-specific critical maximum casting thickness, the critical diameter $d_c$, for which fully amorphous rod-shaped samples can be cast. This size limitation can be overcome by additive manufacturing (AM) techniques, such as LPBF. In this context, the most prominent benefit of AM technology is the possibility to produce large-scale and complex parts through a layer-wise production route, allowing it to decouple part size and cooling rates in the melt pool.

Some authors have obtained a fully glassy state in additively manufactured Fe-based and Zr-based alloys [5,10–17]. These accomplishments open new doors to the investigation of other metallic systems, such as Cu-based metallic glasses. These alloy systems are promising candidates for the AM process due to their high GFA, high tensile strength and elongation, and relative low cost of elements [18–22]. Although their high GFA permits achieving critical casting thickness above 20 mm by copper mold injection casting [23,24], they are restricted to simple geometries and specific alloying compositions, which limit the vast usage of Cu-based BMGs in industrial applications as engineering materials when produced by casting.

The knowledge of additive manufacturing for the production of metallic glasses is in its early stages. Commonly, porosity, cracks, and increased surface roughness are present in the AM parts [11,13,14,16] and can influence the typically reported high corrosion resistance of metallic glasses [25]. The complex thermal cycles in the melt pools and heat-affected zones (HAZs) inherent from consecutive laser scanning mostly generate unique microstructures with chemical heterogeneities, in which commonly partially nanocrystalline phases are present in BMGs [13,14,26,27]. Furthermore, the gas atomization process increases the oxygen content of particles due to the large surface-to-volume ratio, which enhances incorporation of oxygen on the surface [17,28]. Moreover, the extra melting step (melting of master alloy for powder production and consecutively melting with laser scanning) enhances the intake of oxygen. Commonly, the AM-BMGs contain a much higher oxygen content than their cast counterparts [12,14,29], which may be extremely harmful to the resistance of corrosion as it favors the formation of oxides. The corrosion resistance and reactivity of metallic glass produced by additive manufacturing techniques should be adequate to the corresponding application and, therefore, validated.

In this study, the surface reactivity of $Cu_{47}Ti_{33}Zr_{11}Ni_6Sn_2Si_1$ (at. %) metallic glass is investigated as a function of the processing route, i.e., LPBF-formed and conventionally cast samples are directly compared. The $Cu_{47}Ti_{33}Zr_{11}Ni_6Sn_2Si_1$ alloy was selected due to its high glass-forming ability [30] and high yield strength [31]. This alloy composition has not been additively manufactured before. Both samples were investigated in terms of oxygen content by hot gas extraction, crystalline content with X-ray diffraction (XRD), chemical distribution of elements using electron microprobe (EMP), and surface alteration before and after immersion in a corrosive solution (concentrated HCl) based on vertical scanning interferometry (VSI) [32]. This microscopic technique has been successfully used to investigate the surface reactivity of minerals and is being applied for the first time to

BMG systems. The aim of this work is to elucidate the differences between each technique and improve the usage of LPBF as an alternative route for BMG production.

## 2. Materials and Methods

### 2.1. Material Processing and Characterization

To process the as-cast material, master alloys of $Cu_{47}Ti_{33}Zr_{11}Ni_6Sn_2Si_1$ (at. %) were prepared by melting together high-purity elements (purity > 99.9%) in a custom-built arc-melter in a Ti-gettered argon atmosphere. During this procedure, the master alloys were remelted several times for homogenization purposes. In the second step, the alloy was arc-melted under a Ti-gettered argon atmosphere and cast in a custom-built arc-melt suction-casting apparatus into a water-cooled Cu mold with the plate-shaped dimensions of $2 \times 13 \times 50$ mm. X-ray diffraction (PANalytical X'Pert Pro diffractometer, Malvern Panalytical, Malvern, UK) confirmed the formation of the amorphous state within the detection limit of the method.

The powder was synthesized in a close-coupled gas atomizer (CCA) [33] from Indutherm, Walzbachtal, Germany, at the Leibniz-IWT Institute (atomization number PA5-762). The feedstock material consisted of high purity, industrial-grade elements Cu, Ti, Zr, Ni, and Sn and a pre-alloy $Si_{21}Ni_{79}$ (at. %) prepared by arc-melting as described above. They were mixed and placed inside the crucible for melting. A high purity graphite crucible purified under halogen gas flow (grade 2120, Mersen, Courbevoie, France) was selected to minimize impurities. The process chamber was evacuated to an oxygen content <200 ppm and subsequently supplied with argon as process gas to minimize the contamination with oxygen. The melt was heated to a maximum temperature of 1350 °C and held for 7 min to assure homogenization. The processing parameters were selected based on Refs. [33–35]. The resulting gas-atomized powder was classified with standard sieves between 20 and 90 μm to be used in LPBF. The particle shape factors were measured by the static method (Morphologi G3, Malvern Panalytical, Malvern, UK). The Hall Flowmeter funnel with a 2.5 mm nozzle (NanoTech Industrie Produkte, Berlin, Germany) assessed the powder flowability according to DIN EN ISO 4490:2008, static start to flow.

Test blocks of dimensions $(4 \times 4 \times 8)$ mm$^3$ were produced by means of LPBF (Aconity GMBH, Aconity 3D Mini, Herzogenrath, Germany). The longest dimension corresponds to the built direction. A built platform of stainless steel 316L with a 55 mm diameter was employed. To scan for a most-appropriate parameter window, several samples with variated fabrication parameters were created; the range of the applied parameters is shown in Table 1.

**Table 1.** LPBF (laser powder bed fusion) process parameters.

| Laser Power (P) | Scanning Speed (v) | Spot Size | Hatch Distance (h) |
|---|---|---|---|
| 100–400 W | 250–1250 mm s$^{-1}$ | 80 μm | 100/200 μm |

| Layer Thickness | Rotation | Gas | |
|---|---|---|---|
| 50 μm | 67° | argon | |

The amorphous state of powder and LPBF specimens were assessed by XRD (Cu-K$\alpha$ radiation, Bruker D8 ADVANCE diffractometer, Billerica, MA, USA) with a step size of 0.0148°. The X'Pert HighScore software (version 3.0, Malvern Panalytical, Malvern, UK) for powder pattern analysis was used for phase identification. The oxygen content of powder, cast, and LPBF specimens was measured for a sample of approx. 50 mg mass with hot gas extraction (ELTRA ONH-2000, Eltra, Haan, Germany) using helium (99.999%) as the carrier gas and Ni capsules initially cleaned with acetone and dried on a heating plate.

### 2.2. Corrosion Experimental Methods

The samples were embedded in epoxy and mechanically polished with micro-diamond polishing suspension down to 0.25 μm for surface investigation. As described in Ref. [36],

an inert mask was applied to a small area of the surface to protect it against corrosion and serve as the height reference (h = 0) for the topography maps. The samples were immersed in 0.1 M HCl solutions prepared with 37% HCl (12 mol L$^{-1}$) and Milli-Q water (resistivity >18.2 MΩ cm) for a total of 21 days (i.e., 504 h). The batch experiment was conducted at room temperature (21 °C) and in laboratory air for the initial 14 days. Afterward, the experiment was continued in a conventional oven at 40 °C for three days, and sequentially at 60 °C until the final reaction time. After the experiment, the samples were cleaned with ethanol and dried with high-pressure air directly before each VSI measurement. A VSI from Bruker ContourGT (Billerica, MA, USA) equipped with a 5× Mirau objective was utilized to generate the topography maps before and after the experiment, similar to the process described in Refs. [32,37]. The stitching tool integrated into the software Vision64® (version 5.41, Bruker, Billerica, MA, USA) provided broad coverage of the samples. In addition, the software SPIP (version 6.3.5, Image Metrology A/S, Hørsholm, Denmark) was utilized for processing and analyzing the topography images.

The surfaces after corrosion were evaluated with scanning electron microscopy (SEM, JSM-6510, Jeol, Tokyo, Japan). Elemental mapping of O, Cu, Ti, Zr, Ni, Sn, and Si, measured with an electron microprobe (EMP, JXA8200, Jeol, Tokyo, Japan) quantified the chemical composition and distribution on the surfaces. The EMP maps were obtained for an area of 30 × 50 μm using a step size of 0.06 μm, beam current of 20 nA, accelerating voltage of 15 kV, and dwell time of 20 ms. Secondary electron (SE) and backscattered electron (BSE) images were also obtained for each sample with EMP.

## 3. Results

### 3.1. Powder Production and AM-LPBF of Amorphous Samples

Figure 1a, shows the SEM micrograph of the $Cu_{47}Ti_{33}Zr_{11}Ni_6Sn_2Si_1$ powder sieved between 20 and 90 μm. The circularity and aspect ratio measurements resulted in 0.94 and 0.85 on average, and desired flowability for the subsequent LPBF process was obtained, as the average result of three measurements with the Hall flowmeter was 16.6 ± 0.2 s/50 g. During the LPBF process, excessively large melt pools were observed in several specimens, resulting in disturbed heat flow or termination of the process. Additionally, occasional bonding defects occurred between the substrate plate and the test specimen during the manufacturing process. This caused some specimens to detach partially or completely. The sample indicated in Figure 1b, was selected for further investigations in terms of corrosion, surface, and chemical analysis, as it presented the lowest visual flaws. The process parameters of this sample were P = 400 W, v = 1250 mm s$^{-1}$, and h = 200 μm.

To assess the crystallinity, the XRD diffraction patterns of powder, as-cast, and LPBF samples are shown in Figure 1d. They all display the characteristic broad amorphous background. Bragg-peaks connected to crystalline reflections were completely absent in the as-cast specimen and the powder material, which confirmed the formation of an amorphous phase for these two specimens within the detection limit of the method. In contrast, for the LPBF sample, a set of small Bragg-peaks were additionally detected. Due to the relatively weak signal intensity, a clear identification of the phase (either oxide formation of intermetallic compound) was not possible. The detected Bragg-peaks values are listed in Table 2 and seem to correspond to a mixed copper titanium oxide with symmetry Fd3m, either to $Cu_2Ti_4O$ and/or $Cu_3Ti_3O$, as they presented the same symmetry and very similar lattice parameters. The $Cu_2Ti_4O$ phase seemed to be a better match, however.

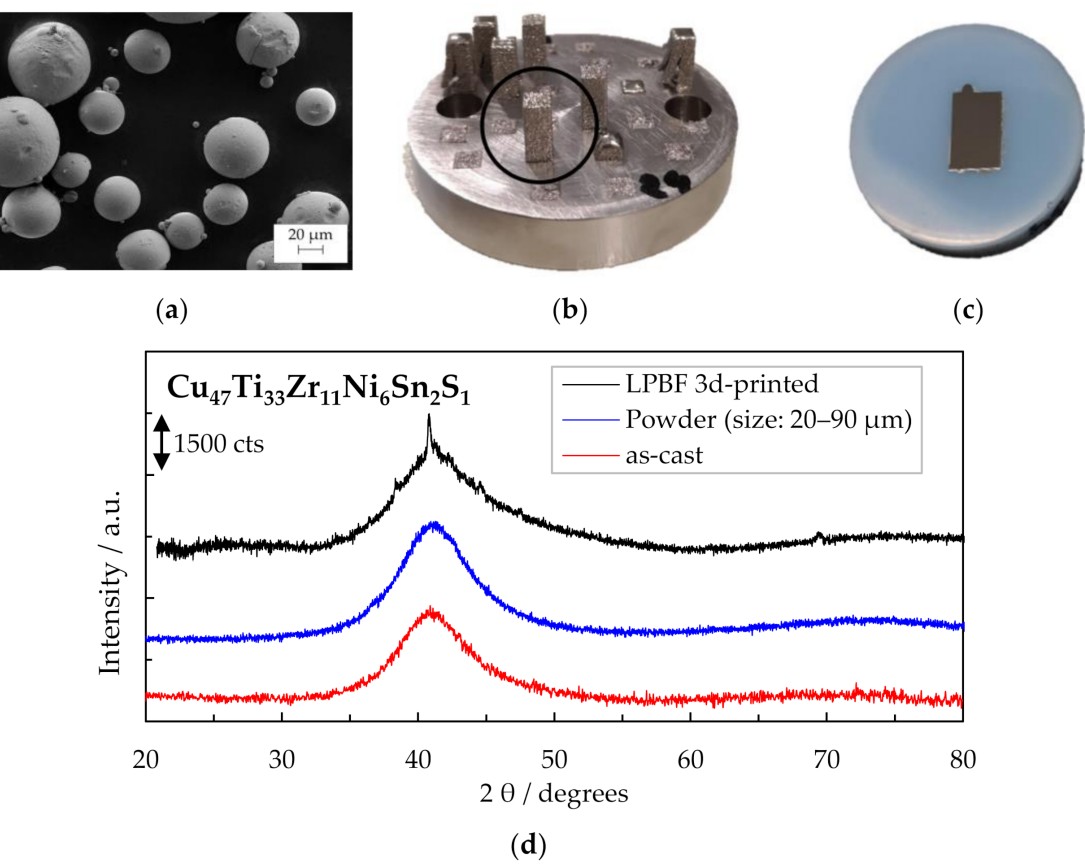

**Figure 1.** (**a**) SEM micrograph of powder as-atomized with a class size of 20–90 μm; (**b**) Photography of LPBF-processes samples, where the sample taken for further analysis is indicated; (**c**) embedded as-cast sample; (**d**) X-ray diffractograms of LPBF, powder, and cast samples. SEM: scanning electron microscopy. LPBF: laser powder bed fusion.

**Table 2.** Bragg-peaks characteristic values detected for the LPBF sample.

| Reflection Position (°2Th.) | Intensity (cts) | Relative Intensity (%) | d-Spacing (Å) |
|---|---|---|---|
| 38.3654 | 282.61 | 20.99 | 2.34626 |
| 40.7804 | 1346.66 | 100.00 | 2.21272 |
| 44.6517 | 182.02 | 13.52 | 2.02946 |
| 69.3603 | 162.62 | 12.08 | 1.35941 |

The oxygen content measured via hot gas extraction of the feedstock powder ($1800 \pm 20$ ppm) was significantly lower than the sample fabricated via LPBF ($3677 \pm 458$ ppm). It appears that, most likely, the residual oxygen in the LPBF chamber was absorbed during the process and incorporated into the material. The reduced oxygen content during arc melting led to a reduced intake of oxygen in the as-cast material ($578 \pm 302$ ppm). The presence of oxygen can affect the GFA of alloys and influence the crystallization behavior [35,38]. The crystallization during the additive manufacturing process was affected by the presence of impurities, including oxides, which acted as heterogeneous nucleation sites [39]. Despite this, a proper heat dissipation within the layers can hinder processing-induced crystallization and enable the freezing of the amorphous phase into a glass; the individual layers are subjected to an accumulated heat input that, in turn, may lead to physical aging and ultimately to crystallization [40,41].

### 3.2. Surface Topography of Samples

Figure 2a,b compares the surface topography of an approx. $300 \times 500$ μm area of the cast sample before and after the immersion in the corrosive media obtained with VSI. A topography line was traced crossing the pre-existing pores on the surface and the height profiles of the initial and final surface are depicted in Figure 2c, with a blue and red

line, respectively. The first valley, located at a distance of approx. 30 μm, represents the deepness of one pore, which reached roughly 250 nm. Despite the apparent decrease in the pore deepness of about 30 nm from the initial to the final topography, no significant difference was seen. The pore size could have restricted the entrance of light during the VSI measurement and could account for such a difference. It is more likely that the pore geometry remained the same, as its walls matched perfectly before and after corrosion. Between the distances of 130 and 150 μm, approximately, a peak followed by a depression of slightly more than 50 nm was seen, which corresponded to the scratch present only at the final surface. The nano-scratch most likely occurred during the handling of the sample between analyses. The second pore was located around 220 μm and the height differences were not statistically significant. Overall, no evidence of the influence of corrosive media was found on the cast material.

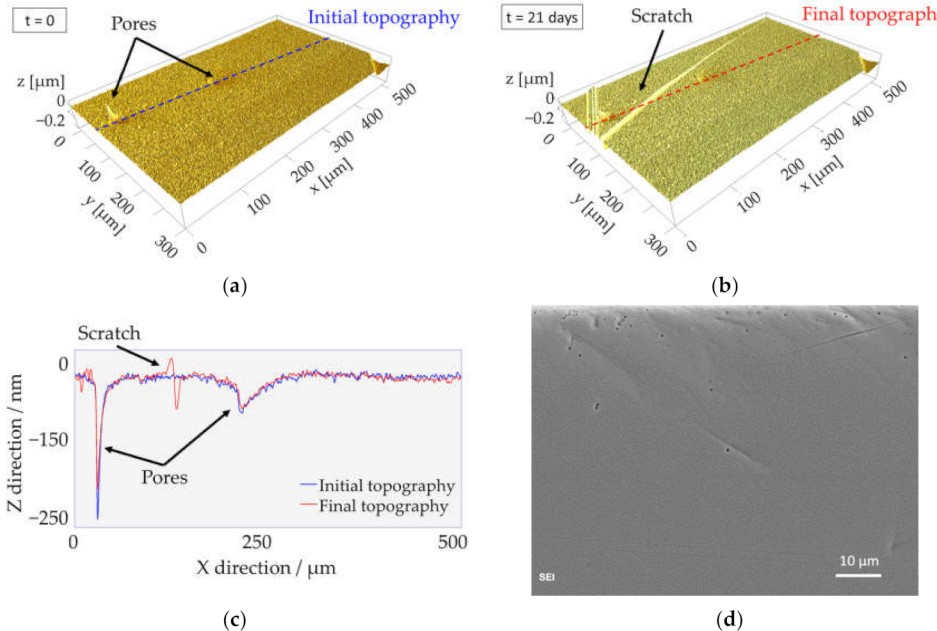

**Figure 2.** Topography analyses for the cast material: (**a**) surface area before corrosion (t = 0) obtained with VSI (vertical scanning interferometry); (**b**) surface post-corrosion (t = 21 days) obtained with VSI; (**c**) the topography line-profiles of the initial (blue curve) and final topography (red curve). The position of the profile is identical and is shown in (**a**,**b**) as a dashed line; (**d**) SEM micrograph of an area near the edge of the sample.

The major concentration of pores in the cast material was seen at the region near the edge of the sample, which is located on the top of the microphotograph in Figure 2d. The edges of the sample were previously in contact with the walls of the mold during casting. It is known that contact with external surfaces such as mold walls may act as potential sites for pore creation due to solidification-related segregation or release of entrapped gas [42]. Our study failed to account for a comparative SEM image preceding corrosion and thus is unsatisfactory to immediately state if these pores were formed due to the immersion in HCl or were created during the manufacturing process. Nevertheless, the results for the elemental distribution at this region, presented in Section 3.3., evidences no chemical variation, corroborating the assumption of porosity defects from processing, as the chemistry inside and around pitted regions usually differ [25,43].

Figure 3 shows the surface topography analysis for the LPBF sample. As revealed by the surface topography at t = 0 (Figure 3a), measured with the VSI, defects of porosity, cracks, and other surface flaws were present, generated during the additive process. These defects are common and associated with the energy input during the manufacturing process, which can be optimized by a better selection of the process parameters towards

denser samples. The subsequent topography after 21 days of immersion in HCl (Figure 3b) clearly shows the propagation of one major crack parallel to the *y*-axis, as indicated by the arrow. The surface height differences are depicted by the profile lines seen in Figure 3c. The blue line indicates the surface roughness before corrosion whereas the red line indicates the final topography, after corrosion. It seems as if the surface planes generated by the crack propagation had moved opposite in upward and downward directions. The large step in the red curve at a distance of approx. 125 µm is connected to this height difference and the associated elevation discrepancy reaches up to 3 µm. Contrary to the result presented for the cast material, the topography lines in the LPBF sample are not traced crossing the pores present in the surface. During the VSI measurement, insufficient light was captured by the detector at these points (improper reflectance), leading to void pixels on the surface in which no information of height is present. Therefore, further data collection is required to determine the exact deepness of the pores. The SEM micrograph after corrosion is seen in Figure 3d. The area inside the traced lines indicates the measured surface topographies with VSI seen in Figure 3a,b and reveals the aforementioned crack propagation detected after the corrosion experiment. This propagation is a result of the expansion of a much bigger crack. This major crack and other defects present in the SEM micrograph probably account for defects generated during the AM process, although a comparative image of the as-prepared LPBF sample is not available. The surface deterioration after the corrosion experiment is linked to the crack propagation detected with VSI and implies a higher surface reactivity generated by the LPBF process in comparison to the cast sample, in which no surface alteration or roughness increase was observed.

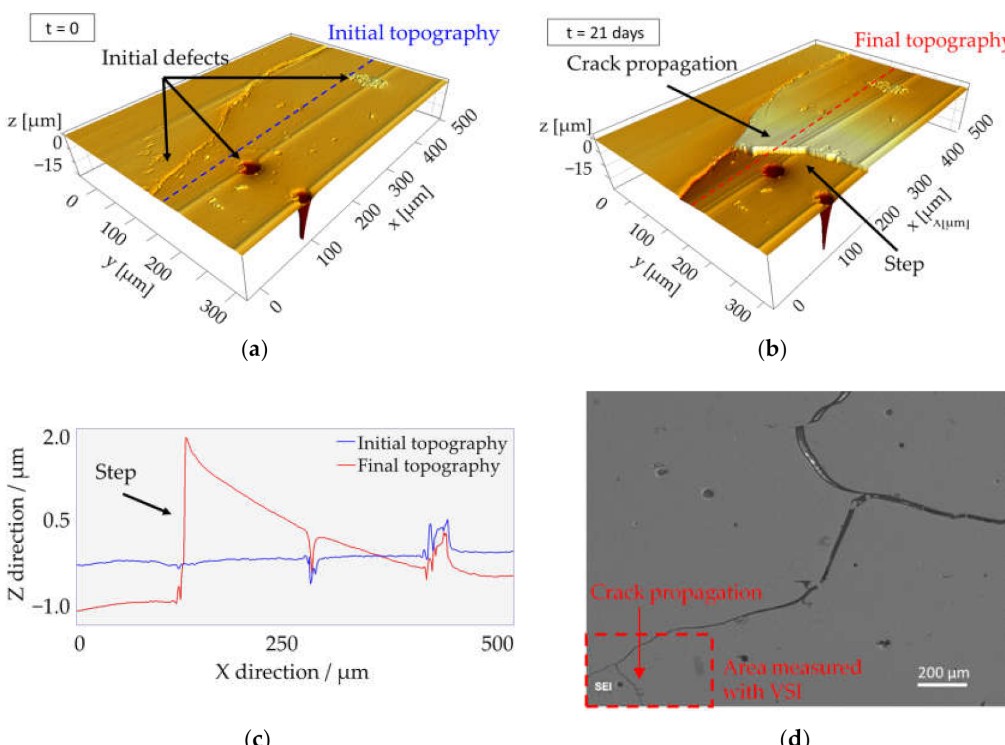

**Figure 3.** Topography analyses for the LPBF material: (**a**) surface area before corrosion (t = 0) obtained with VSI; (**b**) surface post-corrosion (t = 21 days) obtained with VSI; (**c**) The topography line-profiles of the initial (blue curve) and final topography (red curve). The position of the profile is identical and is shown in (**a**,**b**) as a dashed line; (**d**) SEM micrograph of a major crack present in the surface, indicating the area measured with the VSI.

### 3.3. Surface Chemistry of Samples

Figure 4 shows the micrographs obtained with secondary and backscattered electrons (BSE), and the elemental chemical distributions of O, Cu, Ti, Zr, Ni, Sn, and Si for the cast

sample after the dissolution experiment. The mapping was acquired at a region near the edge of the sample, as so to visualize possible chemical alterations in the area where pores were observed (Figure 2d). The top of the images corresponds to the border of the sample. The edges of the sample seem to be in a different focus, maybe caused by polishing, which interferes with the chemical analyses at this region. Nonetheless, the analysis did not show any significant differences in the distribution of elements within the detection limit of the equipment; it revealed a strong homogeneity of the main elements and oxygen. This is in good agreement with the homogeneous single-phase expected for BMGs produced by casting. This result has further strengthened the hypothesis that the pores were a result of the casting process rather than corrosion pits.

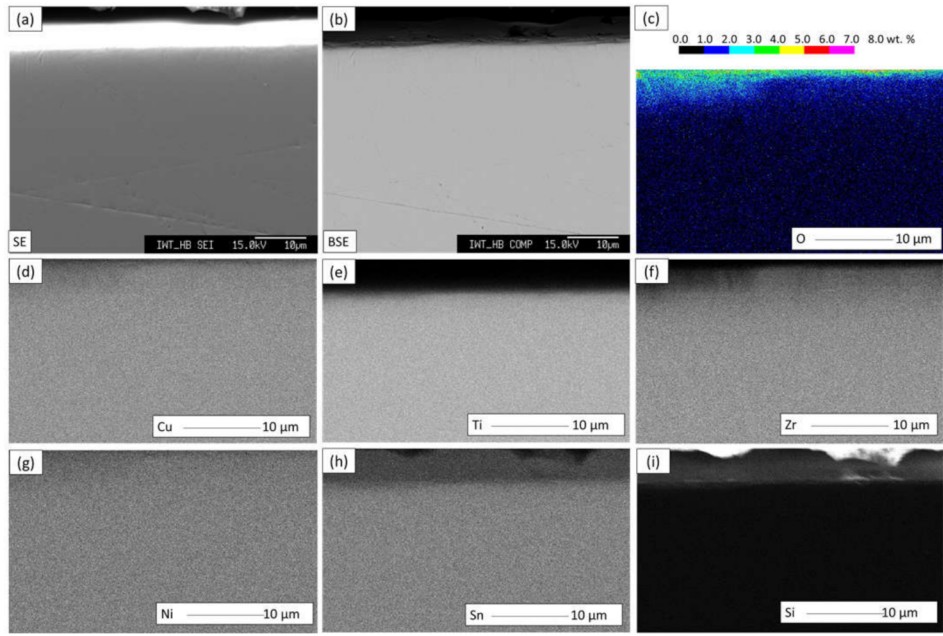

**Figure 4.** EMP (electron microprobe) results of the cast sample after the corrosion investigation including: (**a**) SE (secondary electron micrograph); (**b**) BSE (backscattered electron) micrograph; and the elemental mapping distribution of (**c**) O, (**d**) Cu, (**e**) Ti, (**f**) Zr, (**g**) Ni, (**h**) Sn, and (**i**) Si.

In contrast, Figure 5 exhibits variations in the chemical elemental distribution of the LPBF sample and reveals the influence of the processing route on the microstructure. The surface topography was revealed with the SE micrography (Figure 5a) and the backscattered electron detector (Figure 5b) exposed variations in relative density. The layered structure of the element distribution strongly resembled melt pools and heat-affected zones. This supports previous findings in the literature [26,27]. As indicated by the arrow in Figure 5a, there was a preferential area for the formation of precipitates, which coincided with the region richer in oxygen (Area 1, Figure 5c). The higher concentrations of oxygen were accompanied by a higher presence of Ti, whereas the Cu content was slightly inferior. It is plausible that these precipitates can be assimilated to the crystalline phase $Cu_2Ti_4O$ previously detected with XRD (Figure 1d), as the phase was more enriched in Ti than Cu. The region indicated by Area 2, however, was enriched in Cu and presents a lower concentration of Ti. The standard electrode potentials of Cu and Ti were +0.58 V ($Cu/Cu^{2+}$) and −1.39 V ($Ti/Ti^{2+}$), respectively [43]. It seems that in regions where the less noble element Ti is present, there is a higher likelihood of electrochemical oxidization. Contrariwise, when the more noble element Cu enriched the region, higher resistance to corrosion initiation was observed. Similar mechanisms have been proposed by Baca et al. [43] to justify Zr and Ti oxides formation and pits enriched in Cu in a melt-spun Vitreloy 101 alloy ($Cu_{47}Ti_{34}Zr_{11}Ni_8$) after immersion in a chloride-containing solution, suggesting that the same mechanism could be present in the LPBF sample. The utmost concentration of

oxygen was detected at the edges of the sample, as indicated by arrows in Figure 5c, which occurred simultaneously to the highest concentration of Zr. Although no direct Zr-oxide formation was detected with XRD, there is a possibility of local crystalline formations.

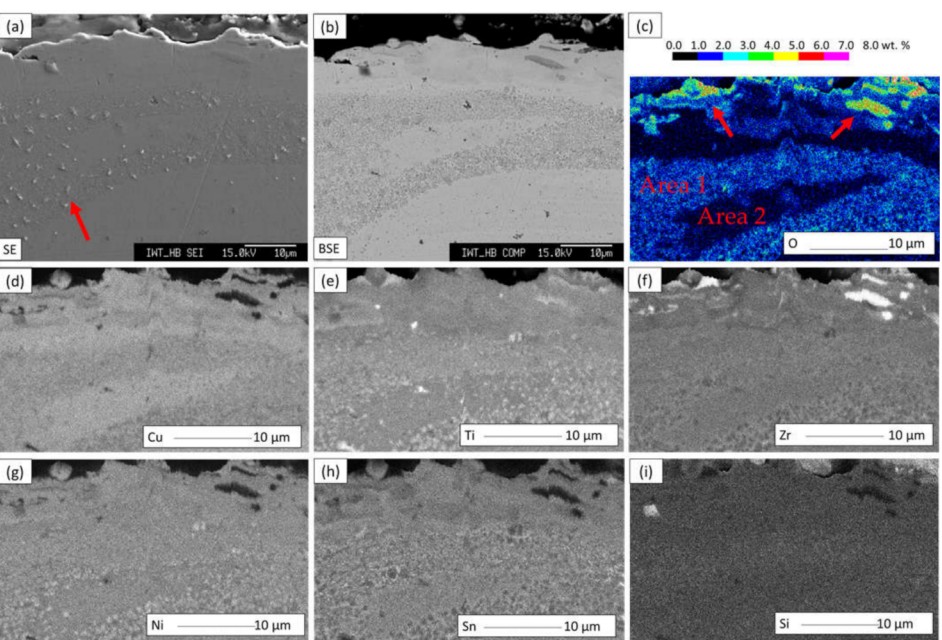

**Figure 5.** EMP results of the LPBF sample after the corrosion investigation including: (**a**) SE micrograph; (**b**) BSE micrograph; and the elemental mapping distribution of (**c**) O, (**d**) Cu, (**e**) Ti, (**f**) Zr, (**g**) Ni, (**h**) Sn, and (**i**) Si. The arrow in (**a**) indicates a preferential area for the formation of precipitates. It occurs at an oxygen-rich region (area 1) as seen in (**c**). The arrows in (**c**) point out the highest concentration of oxygen detected, which coincides with the highest concentration of Zr, as seen in (**f**), suggesting the possible formation of local crystalline formations.

## 4. Discussion

### 4.1. Fabrication Route Implications on the Oxygen Intake and Crystallization

In this study, $Cu_{47}Ti_{33}Zr_{11}Ni_6Sn_2Si_1$ samples were manufactured by mold casting and laser powder bed fusion. The melting and solidification in the casting method occurred in a closed chamber with a Ti-gettered Ar atmosphere. The chances of oxygen contamination are drastically reduced by this cleaning process. This was revealed by the small oxygen content in the cast material (578 $\pm$ 302 ppm). In contrast, during the LPBF process, additional melting steps were necessary, increasing the possibilities of oxygen absorption and other contaminations. First, oxygen can be incorporated during powder production. Although the melting was performed in an inert and evacuated atmosphere, the atomization tower had a much higher volume than the casting mold and it was impossible to completely eliminate the presence of oxygen. Second, the multiple melting during the layer-by-layer process of the LPBF techniques led to more incorporation of oxygen in the fabricated parts. The result from this study showed an oxygen level of 1800 $\pm$ 20 ppm in the feedstock powder (20–90 µm) and 3677 $\pm$ 458 ppm in the LPBF sample, which was almost six times higher than in the cast sample.

The oxygen content played a role in the crystallization of BMGs and, in most cases, an increased content has a detrimental influence on the glass-forming ability [8,14,31]. In turn, as reported in Refs. [8,44], the impairment of the amorphous structure was already sufficient to negatively affect the high corrosion resistance of metallic glasses. It cannot be ruled out that the higher presence of oxygen could have led to the precipitation of crystalline phases. Nevertheless, it should be considered that the reason a fully amorphous phase was not obtained in the LPBF sample is linked to the non-optimized selection of processing parameters. Wegner et al. [16] showed that the energy input derived from the LPBF process

parameters strongly affects the crystallization of BMGs. More recently, Wegner et al. [17] reported the stronger influence of the laser power in comparison with the scan speed on the crystallization of a Zr-based metallic glass during LPBF. Pauly et al. [29] observed for a Zr-based BMG that up to a certain energy density level, fully amorphous samples are obtained based on XRD and differential scanning calorimetry (DSC) observations; however, the increase of such energy density led to a constant rise in the crystalline volume fraction. They also demonstrated that a high laser power improves the relative density until a certain point [29]. Due to the high GFA of the $Cu_{47}Ti_{33}Zr_{11}Ni_6Sn_2Si_1$ and given that cooling rates in the LPBF route are higher than $10^4$ K s$^{-1}$ [14,29], it was expected that a fully amorphous state could be obtained for this alloy by a careful parameter selection, despite the higher oxygen content.

Future research should look for longer reaction times and other corrosive solutions aiming to intentionally corrode the surface of the samples, i.e., promote the occurrence of general corrosion or locally corroded sites (pits). This will enhance the VSI data acquisition and permit the quantification and prediction of corrosion rates and their variability on the surface [36], which is the most advantageous analysis that can be extracted from VSI data for corrosion studies.

### 4.2. Influence of Defects and Chemical Distribution on the Surface Reactivity

A higher number of defects were introduced into the system when using LPBF as the processing route for the Cu-Ti-based metallic glass used in the study in comparison to casting. It has been demonstrated that surface defects weaken the corrosion resistance of alloys to the extent of acting as preferential sites for pit initiation and increasing corrosion susceptibility [25]. Regardless, the presence of pores and cracks can be partially reduced by a proper parameter selection as it became evident that the corrosive solution had a greater impact on the surface of the LPBF sample than the cast material and seemed to have triggered crack propagation as a mechanism of stress release. A similar observation was reported by Baca et al. [43]. The group investigated the corrosion in concentrated HCl of a spun-ribbon Vitreloy 101 alloy ($Cu_{47}Ti_{34}Zr_{11}Ni_8$), thus a similar alloy used in the present study, and they reported the initiation and propagation of large cracks mostly near pitting corrosion sites. Although not confirmed by the authors, they argued that the cracks could be the release of residual stress incorporated during melt spinning by corrosive processes. Residual stress can be accumulated into additively manufactured parts due to the unique thermal history of the process, which is characterized by cyclical large thermal gradients arising from rapid heating and fast cooling rates [11,25]. This is characteristic of the layered additive process and cannot be surpassed. Additional work can target the elemental distribution near cracks formed during the additive manufacturing process and after the corrosion experiment, as well as compare them to EMP maps of as-prepared LPBF specimens. This will fundamentally contribute to understanding the mechanisms of crack formation and propagation in AM-BMGs and benefit the development of more corrosion-resistant parts.

The complex thermal process conditions present in the LPBF have also influenced the microstructure and chemical distribution of elements. It is known that the solidification of the melt pool commonly induces the formation of unique and heterogeneous microstructures [26,27]. Consequently, the performance of the manufactured parts can be affected and may depend on the orientation of the build (anisotropic properties) [12,25]. The heterogeneous distribution of the chemical elements in the LPBF sample shows simultaneously Ti-rich, O-rich, and Cu-depleted areas, where the precipitations seem to favorably occur (Figure 5). Overall, these findings agree with the observations of Ouyang et al. [26], who evaluated the microstructure of the melt pool and HAZ in a 3D-printed Zr-based metallic glass. They recognized the formation of nanocrystals in the Zr-enriched and Cu-depleted HAZ as a result of recurrent laser scanning during the process. Their argumentation can be considered equivalent to our findings, although further investigation is required.

## 5. Conclusions

The present work investigates the influence of the processing route on the surface reactivity of a Cu-based bulk metallic glass based on a comparison between casting and laser powder bed fusion. The unique thermal history of the LPBF process combined with increased melting steps in the process chain, i.e., from powder production until the final manufactured part, have led to an increased oxygen content and partial crystallization of the sample. The elemental distribution suggests that the crystallization favorably occurs in Ti-rich and Cu-depleted regions, where coincidently a higher concentration of oxygen is observed. Presumably, these combined factors affected the surface behavior during corrosive immersion tests in concentrated HCl, where substantial crack propagations associated with stress release occurred and deteriorated the surface. Contrarily, the sample produced by casting had a reasonably lower oxygen content, no major manufacturing defects (as opposed to large pores and cracks present in the LPBF sample), and suffered no alteration at the end of the corrosion study based on vertical scanning interferometry analyses. In addition, it presented a homogeneous chemical distribution, supporting the idea of a single amorphous phase, which is generally the cause of the superior properties of BMGs.

It was demonstrated that a mainly amorphous state can be obtained in additively manufactured $Cu_{47}Ti_{33}Zr_{11}Ni_6Sn_2Si_1$ alloys. Yet, research efforts surrounding laser processing windows toward better control of the melt pool solidification are suggested to fabricate fully amorphous dense parts with fewer defects. Likewise, it is recommended to increase either the reaction time during the corrosion experiments or the aggressiveness of the solution to intentionally detriment the surface of the materials. This will enhance the VSI data acquisition, permit the quantification of corrosion rates based on material dissolution, and help elucidate further influences of the processing route on the surface reactivity of BMGs.

**Author Contributions:** Conceptualization, A.L. and V.U.; methodology, E.S.B., A.L. and V.U.; validation, I.G. and R.B.; formal analysis, E.S.B., I.G., R.B. and M.F.; investigation, E.S.B. and M.F.; resources, V.U. and R.B.; data curation, E.S.B.; writing—original draft preparation, E.S.B.; writing—review and editing, M.F., I.G. and V.U.; visualization, I.G. and R.B.; supervision, A.L. and V.U.; project administration, A.L.; All authors have read and agreed to the published version of the manuscript.

**Funding:** This research received no external funding.

**Institutional Review Board Statement:** Not applicable.

**Informed Consent Statement:** Not applicable.

**Data Availability Statement:** Data sharing is not applicable to this article.

**Acknowledgments:** The authors would like to specially thank the support of Elisabete Trindade Pedrosa, Johannes Birkenstock, Michael Wendschuh, Marco Toro, Petra Witte, Ellen Matthaei-Schulz, Stefanie Schmidt, Martina Rickers, Petra Meier, Frank Peschel, Stefan Evers, and Nico Neuber for their fruitful discussions and assistance with this study.

**Conflicts of Interest:** The authors declare no conflict of interest.

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
