# Peer review of "Influence of Processing Route on the Surface Reactivity of Cu47Ti33Zr11Ni6Sn2Si1 Metallic Glass"

_metals, doi:10.3390/met11081173_

Round 1

Reviewer 1 Report

This work compares the corrosion property of CuTiZrNiSnSi bulk metallic glasses manufactured by copper mold casting and LPBF. They made the conclusion that the a higher oxygen concentration and crystallization degree is achieved in the LPBF sample, leading to the substantial crack propagation after corrosion. This work gives a relatively reasonable analyses of the corrosion behavior of the samples. However, there are some concerns to be checked before it can be published on Metals.

  1. Authors provide elemental mapping of all the elements in Fig. 4 and 5. However, they didn’t state how they acquire the data either in the experimental part or in the results section. Besides, the captions of Fig. 4 and 5 need to be revised. The captions should describe the contents in (a), (b) (c)…. And based on the text, Figs 4 and 5 are not just the elemental mapping but also have the SE, BSE images.

  1. In section 3.1, the last paragraph, how did author get the results of the oxygen concentration in each sample?

  1. The cracks in Fig. 3d, it is difficult to tell whether it is formed during the LPBF process or the corrosion process. An SEM image of the as-prepared LPBF sample is required.

  1. The abbreviations in the manuscript need to be thoroughly checked. For example, in the abstract, LPBF first appear, but no full name of “LPBF” is provided. But in the conclusion part, the “laser powder bed fusion (LPBF)” appear again after LPBF has been mentioned many times in the manuscript.

5. The conclusion of the manuscript is predictable. It is a well-known knowledge, defects, oxygen, elemental segregation are easier to found in LPBF sample than in the copper mold casted sample. And the corrosion resistance of the LPBF sample is worse than the copper mold casted sample. And this manuscript only gives a qualitative analyses, no quantitative analyses of the corrosion behavior is provided. What is the innovation/novelty of this work?  

Author Response

Point 1: Authors provide elemental mapping of all the elements in Fig. 4 and 5. However, they didn’t state how they acquire the data either in the experimental part or in the results section. Besides, the captions of Fig. 4 and 5 need to be revised. The captions should describe the contents in (a), (b) (c)…. And based on the text, Figs 4 and 5 are not just the elemental mapping but also have the SE, BSE images. 

 Response 1: We mentioned at the end of section 2.2 that the elemental mapping was obtained with an electron microprobe after corrosion. Indeed, we agree that information was lacking and, thus, the measurement parameters were added. In addition, we stated that secondary and backscattered electron images were also acquired with the EMP and corrected the captions of Figures 4 and 5.

The surfaces after corrosion were evaluated with scanning electron microscopy (SEM, JSM-6510, Jeol). Elemental mapping of O, Cu, Ti, Zr, Ni, Sn, and Si, measured with an electron microprobe (JXA8200, Jeol) quantified the chemical composition and distribution on the surfaces. The EMP maps were obtained for an area of 30 x 50 µm using a step size of 0.06 µm, beam current of 20 nA, accelerating voltage of 15 kV, and dwell time of 20 ms. Secondary electron (SE) and backscattered electron (BSE) images were also obtained for each sample with EMP.

Point 2: In section 3.1, the last paragraph, how did author get the results of the oxygen concentration in each sample?

Response 2: We informed in section 2.1, just below Table 1, that hot gas extraction was used to measure the oxygen content of powder, cast, and LPBF specimens. We admit that details on the method description were missing and completed the information. Moreover, to remember the reader how the oxygen content was obtained in the result section, we restated that hot gas extraction was used (line 186).

The oxygen content of powder, cast, and LPBF specimens was measured in approx. 50 mg sample with hot gas extraction (ELTRA ONH-2000) using helium (99.999%) as the carrier gas and Ni capsules initially cleaned with acetone and dried on a heating plate.

Point 3: The cracks in Fig. 3d, it is difficult to tell whether it is formed during the LPBF process or the corrosion process. An SEM image of the as-prepared LPBF sample is required.

Response 3: We acknowledge that it is not possible to tell only with Fig. 3d whether the cracks seen in the SEM micrograph are a result of corrosion or of the additive manufacturing process. Unfortunately, an SEM micrograph before corrosion was not obtained. Nevertheless, we can affirm that the crack propagation mentioned in the text (also highlighted inside the traced lines in Fig. 3d) occurred after the immersion in corrosion, as confirmed by the surface topographies provided in Fig.3 a and b. To make it more understandable for the reader, we slightly modified Fig. 3b and d to highlight the crack propagation which occurred after corrosion. We added information along the text (lines 250-259) to address the reviewer’s opinion.

The SEM micrograph after corrosion is seen in Figure 3, d. The area inside the traced lines indicates the measured surface topographies with VSI seen in Figure 3, a and b, and reveals the aforementioned crack propagation detected after the corrosion experiment. It can be seen that this propagation is a result of the expansion of a much bigger crack. This major crack and other defects present in the SEM micrograph probably account for defects generated during the AM process, although a comparative image of the as-prepared LPBF sample is not available. The surface deterioration after the corrosion experiment is linked to the crack propagation detected with VSI and implies a higher surface reactivity generated by the LPBF process in comparison to the cast sample, where no surface alteration or roughness increase was observed.

Point 4: The abbreviations in the manuscript need to be thoroughly checked. For example, in the abstract, LPBF first appear, but no full name of “LPBF” is provided. But in the conclusion part, the “laser powder bed fusion (LPBF)” appear again after LPBF has been mentioned many times in the manuscript.

Response 4: The use of abbreviations was doubled-checked throughout the manuscript.

Point 5: The conclusion of the manuscript is predictable. It is a well-known knowledge, defects, oxygen, elemental segregation are easier to found in LPBF sample than in the copper mold casted sample. And the corrosion resistance of the LPBF sample is worse than the copper mold casted sample. And this manuscript only gives a qualitative analysis, no quantitative analyses of the corrosion behavior is provided. What is the innovation/novelty of this work? 

Response 5: The investigation using VSI has been successfully applied to amorphous solidifying metals and different solidification histories. This proves that this method can be used for other metallic glasses in the future and be extended by quantitative results to better understand the corrosion processes in these materials, as stated in the last paragraph of section 4.1. To address the reviewer’s opinion, we included in the introduction section the information that this alloy is being additively manufactured for the first time, that this microscopic technique hasn’t been used to investigate BMG systems before, and improved our overview for future work in the conclusion section.

      Therefore, it is recommended that further research focus on the parameter optimization of the LPBF process towards better control of the melt pool solidification, besides the fabrication of fully amorphous dense parts with fewer defects to even the comparison between cast and LPBF. Likewise, it is recommended to increase either the reaction time during the corrosion experiments or the aggressiveness of the solution to intentionally detriment the surface of the materials. This would enhance the VSI data acquisition, permit the quantification of corrosion rates based on material dissolution and help elucidate further influences of the processing route on the surface reactivity of BMGs.

Reviewer 2 Report

Effect of additive manufacturing of a Cu-based bulk metallic glass on its surface properties was reported. The incorporation of oxygen during powder production and additive manufacturing processes was well described and its influence on the surface structure and corrosion behavior of the final product was extensively reported. The manuscript is well written and provides valuable information that deserves attention of the researchers in the field. Therefore, publication of the manuscript is recommended. Some minor comments are as follows,

In abstract (line 18), 'chemical inhomogeneous composition' could be replaced by 'compositional inhomogeneity'.

As shown in Figure 3, the LPBF sample produced cracks to release the residual stress after 21 day immersion in HCl solution. However, regarding what initiated such a crack, it is still not clear. Probably, authors may want to provide the elemental mapping information of LPBF sample before corrosion experiment and compare with Figure 5 (after corrosion experiment) to understand/discuss the details of development of surface reaction during the corrosion experiment.

Author Response

Point 1: In abstract (line 18), 'chemical inhomogeneous composition' could be replaced by 'compositional inhomogeneity'.

Response 1: The suggestion of the reviewer was accepted.

Point 2: As shown in Figure 3, the LPBF sample produced cracks to release the residual stress after 21-day immersion in HCl solution. However, regarding what initiated such a crack, it is still not clear. Probably, authors may want to provide the elemental mapping information of LPBF sample before corrosion experiment and compare with Figure 5 (after corrosion experiment) to understand/discuss the details of development of surface reaction during the corrosion experiment.

Response 2: We agree that elemental mapping information of the LPBF sample before corrosion would sustain the discussion regarding the surface reactivity of this sample. However, we failed to provide this comparison and are not able to obtain this result because the sample was sequentially used in other measurements. To address the reviewer’s remark, we further described this issue in lines 250-259 (“track changes” function on) and recommend future researches to closely investigate the chemistry along cracks present in LPBF samples before and after corrosion (lines 366-371). 

The SEM micrograph after corrosion is seen in Figure 3, d. The area inside the traced lines indicates the measured surface topographies with VSI seen in Figure 3, a and b, and reveals the aforementioned crack propagation detected after the corrosion experiment. It can be seen that this propagation is a result of the expansion of a much bigger crack. This major crack and other defects present in the SEM micrograph probably account for defects generated during the AM process, although a comparative image of the as-prepared LPBF sample is not available. The surface deterioration after the corrosion experiment is linked to the crack propagation detected with VSI and implies a higher surface reactivity generated by the LPBF process in comparison to the cast sample, where no surface alteration or roughness increase was observed.

---

Additional work could target the elemental distribution near cracks formed during the additive manufacturing process and after the corrosion experiment, as well as compare them to EMP maps of as-prepared LPBF specimens. This would fundamentally contribute to understanding the mechanisms of crack formation and propagation in AM-BMGs and benefit the development of more corrosion-resistant parts.

Reviewer 3 Report

This work reports the surface topography and chemistry distribution of the as cast BMG and LPBF 3D printed alloys. However the as prepared LPBF samples are crystallized due to the oxidation process, results in the deteriorative of surface reactivity. It's not clear why they choose the CuTiZr-based MG to perform the LPBF printing process, as proved in this work, cast BMG has better properties than the LPBF printed one. While CuTiZr-BMG  has already realized quite big size by conventional method. In the conclusions the authors mentioned that "it is recommended that further research focus on the parameter optimization of LPBF process towards better control of the melt pool solidification...". This research needs to do more work to fulfil the quality of publication in the journal of METALS. 

Round 2

Reviewer 1 Report

The manuscript has been improved according to reviewer's comments. It can be accepted at the current version.

Author Response

Point 1: English language and style are fine/minor spell check required.

Response 1: The manuscript was revised and small corrections have been done. We believe that the English language and style are now appropriate for publication.

Reviewer 3 Report

I have no more comments.